# Effects of Medium Cut-Off Polyarylethersulfone and Polyvinylpyrrolidone Blend Membrane Dialyzers in Hemodialysis Patients: A Systematic Review and Meta-Analysis of Randomized Controlled Trials

**DOI:** 10.3390/membranes12050443

**Published:** 2022-04-20

**Authors:** Yu-Hui Hung, Tai-Shuan Lai, Mohamed Belmouaz, Ya-Chun Tu, Chun-Fu Lai, Shuei-Liong Lin, Yung-Ming Chen

**Affiliations:** 1Renal Division, Department of Internal Medicine, National Taiwan University Hospital and National Taiwan University College of Medicine, Taipei 112201, Taiwan; 113746@ntuh.gov.tw (Y.-H.H.); d99849016@ntu.edu.tw (T.-S.L.); angeltuyachun@gmail.com (Y.-C.T.); linsl@ntu.edu.tw (S.-L.L.); chenym@ntuh.gov.tw (Y.-M.C.); 2Department of Nephrology, Poitiers University Hospital, 86000 Poitiers, France; mohamed.belmouaz@chu-poitiers.fr; 3Graduate Institute of Physiology, National Taiwan University College of Medicine, Taipei 112201, Taiwan; 4Department of Geriatrics and Gerontology, National Taiwan University Hospital and National Taiwan University College of Medicine, Taipei 112201, Taiwan

**Keywords:** artificial kidney, albumin, dialysis membrane expanded hemodialysis, beta 2-microglobulin, high retention onset, membrane characterization molecular weight, uremic toxin

## Abstract

The use of medium cut-off (MCO) polyarylethersulfone and polyvinylpyrrolidone blend membrane is an emerging mode in hemodialysis. Recent studies have shown that MCO membranes exhibit a middle high molecular weight uremic toxin clearance superior to standard high flux hemodialysis. We conducted a systematic literature review and meta-analysis of randomized controlled trials to investigate whether MCO membranes efficiently increase the reduction ratio of middle molecules, and to explore the potential clinical applications of MCO membranes. We selected articles that compared beta 2-microglobulin (β2M), kappa free light chain (κFLC), lambda free light chain (λFLC), interleukin-6 (IL-6), and albumin levels among patients undergoing hemodialysis. Five randomized studies with 328 patients were included. The meta-analysis demonstrated a significantly higher reduction ratio of serum β2M (*p* < 0.0001), κFLC (*p* < 0.0001), and λFLC (*p* = 0.02) in the MCO group. No significant difference was found in serum IL-6 levels after hemodialysis. Albumin loss was observed in the MCO group (*p* = 0.04). In conclusion, this meta-analysis study demonstrated the MCO membranes’ superior ability to clear β2M, κFLC, and λFLC. Serum albumin loss is an issue and should be monitored. Further studies are expected to identify whether MCO membranes could significantly improve clinical outcomes and overall survival.

## 1. Introduction

Modern dialysis medicine has been committed to increasing the removal of uremic toxins during therapy, improving patients’ quality of life, and reducing mortality in patients with renal failure. However, standard hemodialysis results in limitations that limit the survival and quality of life of the patients [1,2]. Even under maintenance dialysis, patients with end-stage kidney disease (ESKD) are at a significantly elevated risk of cardiovascular events and infections [3,4]. One of the central points identified as a potential modifiable item is the efficiency of uremic toxin clearance by dialysis treatments [5].

Hemodialysis patients experiece a variety of bothersome clinical symptoms, such as carpal tunnel syndrome [6,7], restless leg syndrome [8,9], pruritus [8,10], poor appetite [8,11,12], anemia [13], and insomnia [14]. Some of these symptoms can be explained by the incomplete removal of middle high molecular weight uremic toxins during hemodialysis [15]. For example, the removal of alpha 1-microglobulin was shown to be an effective therapeutic strategy for restless leg syndrome in dialysis patients [16]. The accumulation of beta 2-microglobulin (molecular weight 11.8 kilodaltons [kDa]) is also associated with restless leg syndrome, carpal tunnel syndrome, and amyloidosis [16,17]. Immunoglobulin light chains, such as kappa free light chain (22 kDa) and lambda free light chain (42 kDa), are middle molecule uremic toxins that interact with B lymphocytes and lead to the activation of transmembrane signaling [18]. The impairment of neutrophil function can also contribute to chronic inflammation leading to increased infection and cardiovascular risks [18]. Chronic inflammation has been related to mortality in hemodialysis patients [19]. Elevated levels of interleukin-6 (21–28 kDa), one of the middle molecule uremic toxins, have been reported in patients with ESKD [19]. Interleukin-6 is an independent predictor of mortality, and therefore the clearance of this toxin could affect the outcomes of dialysis patients [20,21,22,23].

Currently, applying hemodiafiltration (HDF) is a common method which enhances the removal of larger molecular weight uremic toxins [24]. There was a higher reduction ratio with hemodiafiltration for middle uremic toxins such as cystatin C [25], alpha 1-microglobulin [26], and beta 2-microglobulin [27]. However, HDF requires additional equipment, a significant amount of ultrapure replacement fluid, and more advanced training of medical staff. These requirements and costs repressed the generalization of HDF in current hemodialysis practice [8]. With the advances in dialysis medicine, a class of dialyzer composed of a medium cut-off (MCO) membrane has recently emerged. The MCO membrane provides a higher retention onset and molecular weight cut-off (MWCO). The tailored pore sizes of the MCO membranes provide for the removal of middle high uremic toxins [28]. The MWCO of the MCO membrane is slightly lower than that of albumin. Therefore, MWCO exhibits an advantage in preventing the loss of albumin during treatment compared with high cut-off membranes [29,30]. In addition, dialysis with MCO dialyzers provides additional convection during hemodialysis via significant internal filtration without the need for replacement fluid [30]. In 2017, this emerging hemodialysis with MCO membranes was named “expanded hemodialysis (HDx)” [30].

Hypothetically, the increased clearance of middle high molecular weight uremic toxins might improve the clinical symptoms of hemodialysis patients and provide clinical benefits. Recent studies have shown that the MCO membrane dialyzer exhibits a clearance of middle high molecular weight uremic toxins superior to standard high-flux hemodialysis [1,3,31,32,33,34,35,36]. However, most studies included only small numbers of study patients, and not all of them were randomized controlled trials. Therefore, the primary aim of this systematic review and meta-analysis study was to determine whether an MCO membrane could improve the clearance of several middle uremic molecules and improve clinical outcomes.

## 2. Materials and Methods

This meta-analysis was conducted according to the recommendations of the Cochrane collaboration, following the Preferred Reporting Items for Systematic Reviews and Meta-Analyses 2020 guidelines (http://www.prisma-statement.org/PRISMAStatement/CitingAndUsingPRISMA; accessed on 11 January 2021), and was registered in the PROSPERO registry for prospectively registered systematic reviews (registration number: CRD42022307718). The main study population comprised patients with ESKD. The intervention of the meta-analysis used an MCO-polyvinylpyrrolidone blend membrane dialyzer during hemodialysis, compared with high-flux hemodialysis. The measured outcome was the clinical improvement in the clearance of middle molecular weight molecules resulting in an attenuation of the clinical symptoms.

### 2.1. Search Strategy

We performed a comprehensive literature search in several databases from 1 January 2000 to 31 October 2021, including PubMed, EMBASE, the Cochrane central register of controlled trials, and the ClinicalTrials.gov registry (https://clinicaltrials.gov/; accessed on 11 January 2021). For Embase and PubMed, the controlled text search terms were as follows: “Expanded hemodialysis OR HDx OR medium cut-off membrane OR medium cut-off polyvinylpyrrolidone blend membrane OR MCO OR high retention onset membrane” AND “beta-2 microglobulin OR kappa free light chain OR lambda free light chain OR interleukin-6 OR albumin.” We employed related citations in the PubMed search tool to broaden each search, and reviewed all retrieved abstracts, study reports, and related citations. The MeSH search terms in the search strategy are listed in Appendix A. No language restrictions were imposed.

### 2.2. Eligibility Criteria Intervention Comparison

The inclusion criteria were as follows: (1) an RCT, (2) application of an MCO membrane with a dialyzer using an MCO membrane dialyzer or a high retention onset membrane, and (3) a report of the serum levels of middle molecular weight uremic toxins before and after the hemodialysis, including beta 2-microglobulin, kappa free light chain, lambda free light chain, interleukin-6, and albumin. Studies were excluded if they met the following exclusion criteria: (1) not randomized, (2) incomplete trials or data, or (3) use of hemodiafiltration as the control condition instead of high-flux or low-flux hemodialysis. Two reviewers (YH Hung and CF Lia) screened all titles and abstracts under these criteria.

### 2.3. Data Extraction

Two reviewers (YH Hung and CF Lai) extracted the basic characteristics of the studies and outcomes, including the study design, patient sex, inclusion and exclusion criteria, type of MCO dialyzer, the dialyzer employed for the control group, primary and secondary outcomes, and any related complications. Each reviewer independently conducted a critical appraisal. A third reviewer (TS Lai) rechecked all steps from the search to the article writing and made the final decision if a discrepancy was found.

### 2.4. Risk of Bias Assessment

Two investigators (YH Hung and CF Lai) independently assessed each study’s methodological quality based on the adequacy of the randomization, allocation concealment, blinding and outcome assessors, trial duration, reporting of study withdrawal, performance of an intention-to-treat analysis, and other possible sources of bias [37,38]. If no consensus was found between them, a third reviewer (TS Lai) would make the final decision.

### 2.5. Outcomes and Statistical Analysis

The primary outcomes were the serum levels and reduction ratio of the serum levels of selected middle molecular weight uremic toxins, namely beta 2-microglobulin, kappa free light chain, lambda free light chain, and interleukin-6. The secondary outcomes were the serum albumin levels before and after the various hemodialysis sessions. All data were entered and analyzed using Review Manager (RevMan), version 5.4 (Cochrane Collaboration, Oxford, UK). The mean difference was reported for the continuous outcomes, and the effect sizes of the dichotomous outcomes were reported as risk ratios (RRs). The precision of the effect sizes was based on a 95% condifence interval (CI). A pooled estimate of the RR was computed by the DerSimonian and Laird random-effects model, which provides an appropriate estimate of the average treatment effect when the trials are statistically heterogeneous, and usually yields a relatively large CI, resulting in a more conservative statistical claim.

### 2.6. Additional Analysis

To evaluate the statistical heterogeneity and inconsistency of the treatment effects across the studies, we used the Cochran Q test and I^2^ statistics, respectively. Statistical significance was set at 0.10 for the Cochran Q tests. The proportion of the total outcome variability that was attributable to the variability across the studies was quantified as I^2^. Sensitivity analyses were performed to assess the impact of the study quality on the effect estimates. We used the contour-enhanced funnel plot for testing for significant asymmetry which indicates a possible publication or other bias, as well as whether the areas where studies exist are areas of statistical significance, and whether the areas where studies are potentially missing correspond to areas of low statistical significance.

## 3. Results

### 3.1. Literature Search

Figure 1 shows the process applied to screening and inclusion of RCTs. Our initial search yielded 1256 records published between 1993 and 2021, most (1022) of which were not randomized studies and therefore excluded. Among the remaining 228 studies, 212 were also excluded: 186 were not relevant to MCO membranes, 9 were duplicates, and 17 compared MCO membranes with continuous renal replacement therapy or HDF. After reviewing the full text of the remaining 16 studies, 5 eligible RCTs involving 328 participants fit our inclusion criteria and were selected for the study.

### 3.2. Trial Characteristics

Table 1 lists the five selected studies which were published from 2017 to 2020. Kirsh et al. published their study in January 2017 [35]; Zickler et al. published theirs in January 2017 [34]; Belmouaz et al. published theirs in February 2020 [32]; Lim et al. published theirs in May 2020 [36]; and Weiner et al. published theirs in August 2020 [33]. The sample size ranged from 19 to 172 patients. Three of these studies were from European countries, one from the United States, and the one was from South Korea. The reported language for all the studies was English. All the trials recruited patients with ESKD undergoing maintenance hemodialysis with high-flux dialyzers. Other potential biases in the trial characteristics included different MCO membrane dialyzers in the study group, different dialyzers in the control groups, and different experimental periods. The patient characteristics, selected dialyzer, experimental period, and the primary and secondary endpoints in each of the five trials are listed in Table 1. The MCO membrane dialyzers group included Theranova^®^ 400, Theranova^®^ 500, and MCO-Ci 400^®^. One trial compared three different prototypes of an MCO dialyzer (AA, BB, and CC) with a high-flux dialyzer. In all of the included studies, the MCO group was compared with a control group of hemodialysis with high-flux dialyzers (Table 1). Three of the studies disclosed the allocation generation. All the studies were reported as open-label studies due to the difficulty of blinding in a clinical situation.

### 3.3. Primary Outcomes

Figure 2 describes the primary outcomes. All five RCTs evaluated serum beta 2-microglobulin levels before and after the experiments. The reduction ratio of beta 2-microglobulin could not be obtained in one study [39] because its original data were not available. The reduction ratio of beta 2-microglobulin was significantly higher for the MCO group compared with high-flux hemodialysis (*p* < 0.0001). Next, the reduction ratio for kappa free light chain and lambda free light chain were also apparently decreased after MCO (*p* < 0.0001 and *p* = 0.02 respectively). However, the reduction ratio for interleukin-6 did not differ between the groups (*p* = 0.07) [32,33]. The “intergroup serum” of interleukin-6 was not significantly different before (mean difference, MCO vs. high-flux hemodialysis, 0.23 [−0.32 to 0.77], *p* = 0.42, I^2^ = 0), or after (mean difference, −0.56 [−1.48 to 0.35], *p* = 0.23, I^2^ = 34%) the study treatments.

### 3.4. Secondary Outcomes

One study reported statistically significant serum albumin loss with the MCO membrane dialyzer [39], and one study did not report albumin data [35]. In the other RCTs [32,33,36], the serum albumin levels did not statistically decrease after applying the MCO membrane dialyzer. Our meta-analysis showed that the serum albumin levels before treatment were comparable between the MCO and control groups (Figure 3). Compared with the control groups, however, the MCO group demonstrated significantly lower serum albumin levels after the treatment (mean difference between MCO and high-flux hemodialysis, −0.13 [−0.25 to −0.02], *p* = 0.04, I^2^ = 60%).

### 3.5. Publication Bias

To evaluate the publication bias, the comparison of beta 2-microglobulin levels in the MCO and control groups was plotted against the study’s precision groups using a funnel plot. The right limb was missing in the funnel plot, suggesting a potential publication bias (Appendix A).

### 3.6. Sensitivity Analysis

In the five RCTs, only the study by Zickler et al. employed MCO-Ci 400^®^ instead of Theranova^®^ 400 or Theranova^®^ 500 [34]. If we exclude this study, the albumin levels would show no difference between the two groups (mean difference −0.09 g/L [−0.21, 0.04]; *p* = 0.19).

The other potential biases (e.g., the RCT design [concurrent or cross-over], different dialyzers in the control group [Elisio 21H, Elisio 17H, or Revaclear]) showed no significant differences in the sensitivity analysis. The risk of other biases is summarized in Figure 4.

## 4. Discussion

To date, there has been no RCT that reported clinical outcomes after applying an MCO membrane dialyzer, such as overall mortality and cardiovascular events. Therefore, we systematically analyzed the reduction ratio of important middle molecular weight uremic toxins among these studies. In our meta-analysis, the reduction ratios of beta-2 microglobulin and kappa and lambda free light chains demonstrate that these molecules were removed more in the MCO group than in the control group, an expected result given that the MCO dialyzer was designed to remove middle molecules (mainly beta-2 microglobulin) [39]. The increased heterogeneity in the reduction ratios of beta 2-microglobulin and kappa and lambda free light chains was probably design-related due to the differing experimental periods, the differing underlying dialyzers in the control group, and the differing blood flow settings during hemodialysis. This meta-analysis of RCTs confirmed the clinical effects of MCO membrane dialyzers in terms of their enhanced clearance of certain middle molecular weight uremic toxins.

However, MCO polyarylethersulfone and polyvinylpyrroli-done blend membrane dialyzers might not achieve enhanced clearance of other middle molecular weight toxins compared with high-flux hemodialysis. The two included RCTs that addressed the interleukin-6 clearance of MCO did not provide sufficient evidence due to the small sample size (Figure 2d) [32,33]. Possible explanations include the small number of studies addressing interleukin-6 clearance and the small sample size. Esposito et al. attempted to show the difference between MCO and classical hemodialysis in hemodialysis patients with COVID-19. The results showed that no statistically significant changes were found in circulating interleukin-6, interleukin-8, interleukin-10, soluble toll-like receptor 4, or interferon-gamma between the two groups [40]. In contrast, other studies have reported the effectiveness of removing interleukin-6 with MCO [3,39,41]. The interleukin-6 reduction ratio probably depends not only on the molecular weight and distribution volume, but also on the inflammatory response [40].

There were various MCO dialyzers applied in these RCTs, including MCO-Ci 400^®^, Theranova^®^ 400, and Theranova^®^ 500. In our meta-analysis, the loss of albumin only became insignificant when excluding the study by Zickler et al., which employed MCO-Ci 400^®^ instead of Theranova^®^. The prospective study by Maduell et al. compared different surface areas (1.7 and 2.0 m^2^) of the MCO dialyzer and different blood flows (300, 350, 400, or 450 mL/min) [42]. The results showed no statistically significant intergroup differences in the reduction rate of alpha-1 microglobulin, beta-2 microglobulin, myoglobulin, prolactin, or alpha-1 acid glycoprotein, or albumin loss.

Although the MWCO of the MCO membrane is smaller than the molecular weight of albumin [30], this meta-analysis showed that post-treatment serum albumin levels in the MCO group were significantly lower than those in the control group. Our sensitivity analysis also suggests that the clinical impact of albumin loss by MCO might be related to the treatment duration or to the differenct MCO dialyzers employed. The statistical heterogeneity of albumin levels before and after treatment was associated with the differing periods for accepting MCO dialyzer treatment. By excluding the trial with the shortest experimental period (4 weeks) and the only study using MCO-Ci 400^®^ [34,39], the differences in the post-treatment serum albumin levels became statistically insignificant between the two groups (*p* = 0.19). Ronco et al. reported that the albumin loss was 2–4 g per session in six patients undergoing hemodialysis with the MCO dialyzer. However, their serum albumin concentrations were not significantly changed after applying the MCO membrane dialyzer for 6 months [30]. One possible explanation for this observation is that the patients’ appetite might have been improved in the MCO group, which in turn compensated for the loss of albumin by the MCO membrane [8]. Leptin is a 16 kDa middle molecule, and its removal might be associated with changes in the patients’ appetite [30,43,44]. Maduell et al. conducted a prospective study to investigate the effects of the different surface areas (1.7 and 2.0 m^2^) of MCO dialyzers and different blood flow rates (300, 350, 400, or 450 mL/min) on the clearance of uremic toxins [42]. The results showed no statistically significant intergroup differences over the reduction rate of alpha-1 microglobulin, beta-2 microglobulin, myoglobulin, prolactin, or alpha-1 acid glycoprotein, or albumin loss. Overall, when applying an MCO membrane dialyzer, physicians should keep albumin loss in mind and regularly monitor their patients’ nutritional statuses.

### Study Limitations

This study exhibits limitations. First, the sample sizes of the RCTs included in this meta-analysis were relatively small. Second, the MCO membrane dialyzers differed among the trials and we were unable to identify the effects. Third, the heterogeneity of the studies was high, with an I^2^ ranging from 0% to 100%. Furthermore, there is a lack of RCTs investigating the effects of MCO on clinical symptoms, morbidity, and mortality. A superior clearance of middle molecular weight uremic toxins is not the same as a superior clinical outcome.

## 5. Conclusions

In conclusion, this meta-analysis observed a superior clearance of beta 2-microglobulin and kappa and lambda free light chains by dialyzers equipped with MCO polyarylethersulfone and polyvinylpyrroli-done blend membranes compared with high-flux hemodialysis. An MCO dialyzer can be applied to patients with symptoms related to the accumulation of middle uremic toxins. Lower serum albumin levels after MCO treatment were noted in our meta-analysis and should be monitored clinically. To date, there have been no RCTs that have evaluated the long-term cardiovascular risks, hospitalization rate, or overall mortality. Further high-quality studies on the reduction ratio of albumin and on clinical outcomes are needed.

## Figures and Tables

**Figure 1 membranes-12-00443-f001:**
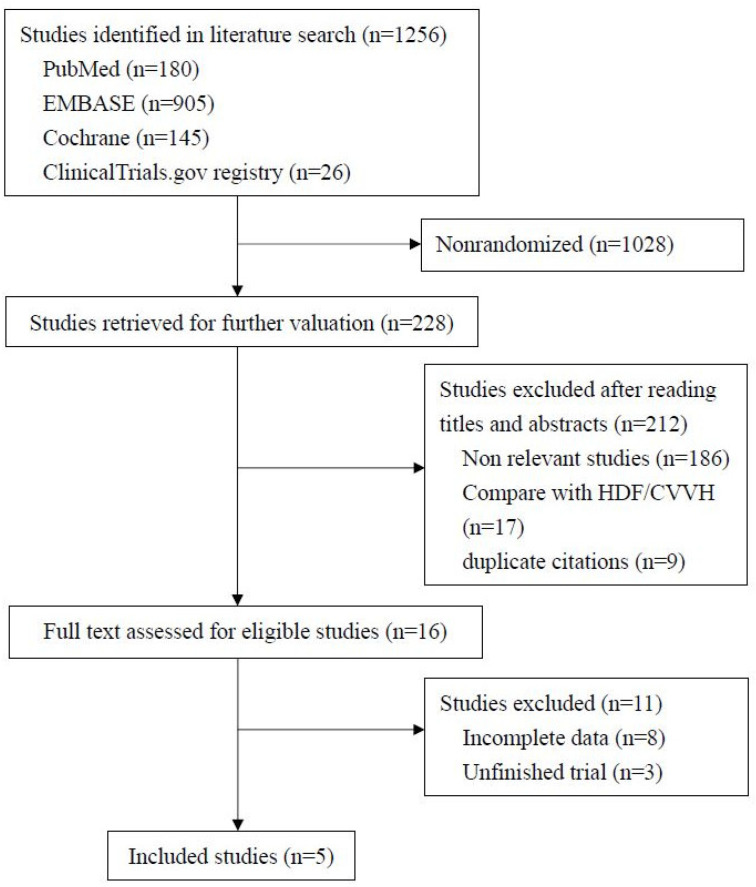
Flow chart for selection of studies.

**Figure 2 membranes-12-00443-f002:**
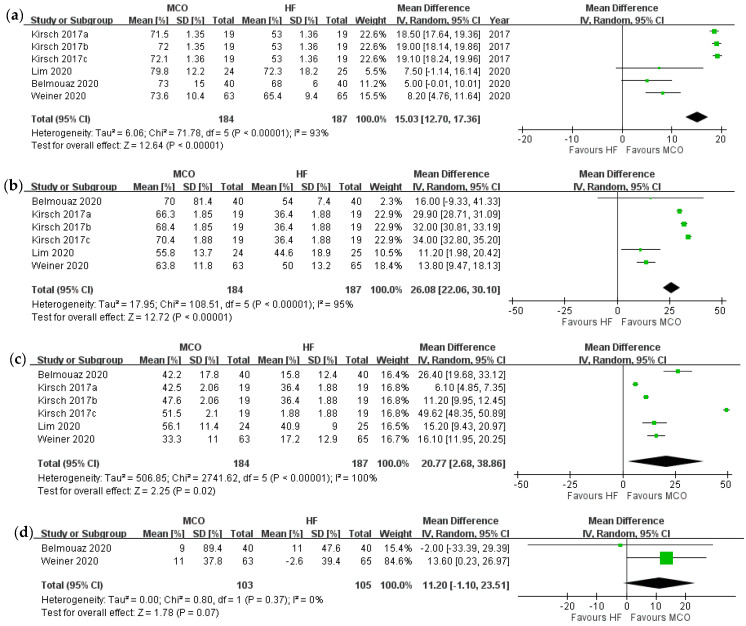
Forrest plot of reduction ratios over different middle molecules: (**a**) reduction ratio of beta 2-mircoglobulin, (**b**) reduction ratio of kappa free light chain, (**c**) reduction ratio of lambda free light chain, (**d**) reduction ratio of interleukin-6. Kirsch 2017a = MCO prototype AA; Kirsch 2017b = MCO prototype BB; Kirsch 2017c = MCO prototype CC.

**Figure 3 membranes-12-00443-f003:**
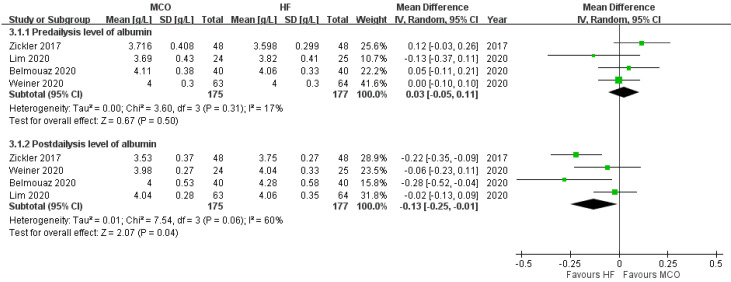
Albumin level before and after hemodialysis.

**Figure 4 membranes-12-00443-f004:**
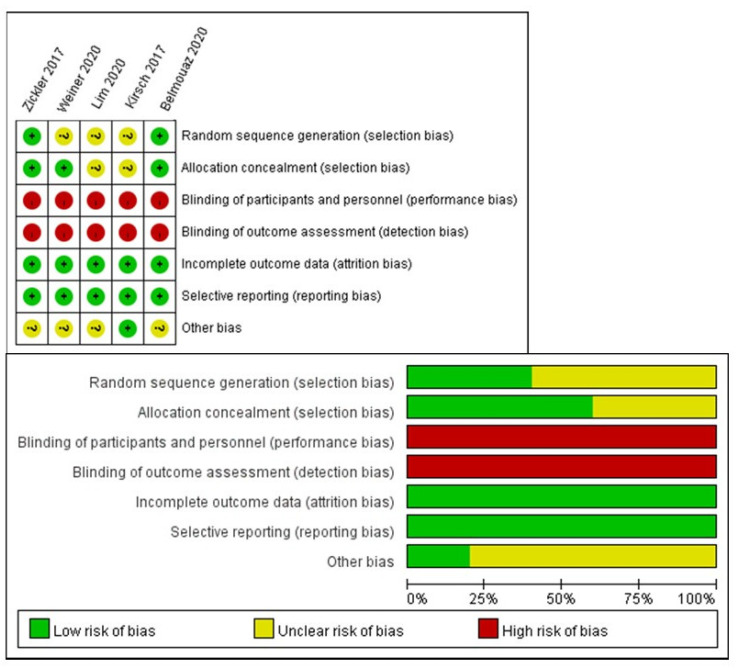
Risk of bias summary.

**Table 1 membranes-12-00443-t001:** Characteristics of the selected randomized controlled trials.

First Author, Year	Country	MCO Membrane Dialyzer	Controlled Group	Method	Experimental Period	Patient Number (Male %)	Primary End-Point	Secondary End-Point
Kirsch 2017	Austria	Theranova^®^400	FX CorDiax 80 FX CorDiax 800 (HDF)	Open-label, cross-over RCT	Once (4–5 h)	19 (72.0%)	kFLC overall clearance	Overall clearances and pre-to-post-reduction ratios of middle and small molecules
Zickler 2018	Germany	MCO-Ci 400^®^	Revaclear 400	Open-label, cross-over RCT	4 weeks of MCO + (4 weeks wash-out) + 4 weeks of high-flux HD	48 (72.9%)	The gene expression of TNF-α and IL-6 in PBMCs	Specified inflammatory mediators, cytokines
Belmouaz 2020	France	Theranova^®^500	Elisio 21H	Open-label, cross-over RCT	3 months of MCO + 3 months of high-flux HD	40 (70.0%)	Myoglobin RR	Other middle-weight toxins
Lim 2020	Korea	Theranova^®^400	FX CorDiax 80 or 60	Open-label, concurrent RCT	12 weeks	MCO: 24 (75%) HF: 25 (60%)	Laboratory data, UF volume, and dialysis adequacy	Middle molecule removal
Weiner 2020	The United States	Theranova^®^400	Elisio-17H	Open-label, concurrent RCT	24 weeks	MCO: 86 (63%) HF: 86 (59%)	Free light chains RR	Complement factor D, free k light chains, TNF-alpha, b2-microglobuli, IL-6

HDF = hemodiafiltration; HF = high flux; kFLC = lambda free light chain; PBMCs = peripheral blood mononuclear cells; RCT = randomized controlled trial; RR = reduction ratio; UF = ultrafiltration.

## Data Availability

Data sharing not applicable.

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
