# Peer review of "Effects of Medium Cut-Off Polyarylethersulfone and Polyvinylpyrrolidone Blend Membrane Dialyzers in Hemodialysis Patients: A Systematic Review and Meta-Analysis of Randomized Controlled Trials"

_membranes, 2022, doi:10.3390/membranes12050443_

Round 1

Reviewer 1 Report

Dear Authors,

Thank you for responding to my remarks comprehensively.

In your objective below, you may want to make a slight changes.

"Therefore, the primary aim of this systemic review and meta-analysis study was to determine whether an MCO membrane could improve the clearance of several middle uremic molecules and affect clinical outcomes’ 

-Suggest change spelling 'systemic' to 'systematic'

-Suggest using the word 'improve' rather than 'affect' clinical outcomes. 

Best wishes

Author Response

Dear Authors,
Thank you for responding to my remarks comprehensively.
In your objective below, you may want to make a slight changes.

We thank the editors and reviewers for the reviews. Our point by point response as follows:

Editor's Comments to Author (Reviewer 1): 
"Therefore, the primary aim of this systemic review and meta-analysis study was to determine whether an MCO membrane could improve the clearance of several middle uremic molecules and affect clinical outcomes’ 

-Suggest change spelling 'systemic' to 'systematic'

Thank you for the suggestion. We corrected the word as your suggested in line 84-86.

“Therefore, the primary aim of this systematic review and meta-analysis study was to determine whether an MCO membrane could improve the clearance of several middle uremic molecules and improve clinical outcomes.”

-Suggest using the word 'improve' rather than 'affect' clinical outcomes. 

Thank you for providing the insight. We revised the last sentence in line 84-86.

“Therefore, the primary aim of this systematic review and meta-analysis study was to determine whether an MCO membrane could improve the clearance of several middle uremic molecules and improve clinical outcomes.”

We thank the reviewer for all these comments.

Reviewer 2 Report

The manuscript is improved

Author Response

We thank the editors and reviewers for the reviews. Our point by point response as follows:

The manuscript is improved

We thank the reviewer for all the comments.

This manuscript is a resubmission of an earlier submission. The following is a list of the peer review reports and author responses from that submission.

Round 1

Reviewer 1 Report

Authors,

Thank you for an opportunity to review this systematic review and meta-analysis that sought to investigate the clinical benefits of MCO membranes in patients on hemodialysis. My remarks are as follows;

-A revision of the paper with special emphasis on the style of writing and grammar will improve this piece of work.

-The last paragraph of the introduction needs to clearly highlight the aim of the study without repetition. Additionally, this is an introduction and should not contain information that needs to be in the methods section for example, ‘We focused on the published RCTs to attenuate potential bias.’

-The opening sentence of the methods section is not clear. Are the authors suggesting that ethical approval was used as criteria to include or exclude studies in this review?

-How different is this sentence (line 91 to 96) from sentence in lines 97 to 99?

-I suggest that the authors clearly highlight a framework that informed their search for instance the PICOs format.

-The search strategy also needs to be provided even as an appendix.

-Please provide the precise dates in which the searched publications were published.

-Regarding data extraction, is the information presented relevant to this particular study? ‘Two reviewers (YH Hung and CF Lai) extracted the baseline and outcome data together, including the study design, the participant data, the inclusion and exclusion  criteria, the anesthetic techniques used, the airway devices used, and any resulting complications.’ This appears to have been extracted from this reference Lam F, Lin YC, Tsai HC, Chen TL, Tam KW, Chen CY. Effect of Intracuff Lidocaine on Postoperative Sore Throat and the Emergence Phenomenon: A Systematic Review and Meta-Analysis of Randomized Controlled Trials. PLoS One. 2015; 10(8):e0136184. Published 2015 Aug 19. doi:10.1371/journal.pone.0136184.

- Was the critical appraisal also performed independently by two reviewers?

-There should be a more detailed description of how risk of bias was assessed. Was it done

independently by one or two or more reviewers? How were disagreements resolved?

-Comment on heterogeneity which appeared to be very high for most of the outcomes. How did this affect the interpretation of your findings?

-Typo in line 230, ‘Until now, there is no RTC of MCO membrane dialyzer reporting patients’ hard…’

-What are the implications of this study to research and clinical practice?

Reviewer 2 Report

Hung  and colleagues did a metanalysis on  the effects of medium cut-Off polyarylethersulfone and polyvinylpyrrolidone blend membrane dialyzers in hemodialysis patients.  The analysis included  five randomized studies with (328 patients), and the authors assessed the efficiency of MCO membrane in the reduction of middle molecules such as β2M-microglobulin , kappa free light chains (κFLC), lambda free light chain (λFLC), IL-6 and albumin among patients . The author concluded that the use of MCO was associated was a significant reduction in β2M-microglobulin, κFLC, and λFLC, but not IL-6.

I have some suggestions on the manuscript in the present form

Major points

1- Some sections in the manuscript are not clear. For example, page 7 lines 218-228 ( Sensitivity analysis), this part is not clear. It is not clear why all the authors need to excluse some studies to get the conclusion.

2- I guess there are some papers misssing from the authors, they need to include

https://www.revistanefrologia.com/en-evaluation-influence-surface-membrane-blood-articulo-S2013251419301506

At least should be included in the discussion

3- The authors need to include int he discussion explaination why MCO did not show signficiant effect on IL-6 and albumin

4- Future diretions section should be also included

Minor comments

1- page 2 lines 89-91: please delete the first paragraph, I guess it a journal template (instruction to authors).